# Single-Cell Mechanophenotyping in Microfluidics to Evaluate Behavior of U87 Glioma Cells

**DOI:** 10.3390/mi11090845

**Published:** 2020-09-11

**Authors:** Esra Sengul, Meltem Elitas

**Affiliations:** 1Faculty of Engineering and Natural Sciences, Sabanci University, 34956 Istanbul, Turkey; sengulesra@sabanciuniv.edu; 2Nanotechnology Research and Application Center, Sabanci University, 34956 Istanbul, Turkey

**Keywords:** glioblastoma, mechanophenotyping, deformation, migration, Label-free, single-cell

## Abstract

Integration of microfabricated, single-cell resolution and traditional, population-level biological assays will be the future of modern techniques in biology that will enroll in the evolution of biology into a precision scientific discipline. In this study, we developed a microfabricated cell culture platform to investigate the indirect influence of macrophages on glioma cell behavior. We quantified proliferation, morphology, motility, migration, and deformation properties of glioma cells at single-cell level and compared these results with population-level data. Our results showed that glioma cells obtained slightly slower proliferation, higher motility, and extremely significant deformation capability when cultured with 50% regular growth medium and 50% macrophage-depleted medium. When the expression levels of E-cadherin and Vimentin proteins were measured, it was verified that observed mechanophenotypic alterations in glioma cells were not due to epithelium to mesenchymal transition. Our results were consistent with previously reported enormous heterogeneity of U87 glioma cell line. Herein, for the first time, we quantified the change of deformation indexes of U87 glioma cells using microfluidic devices for single-cells analysis.

## 1. Introduction

Glioblastoma or Glioblastoma multiforme (GBM), is the most devastating type of brain cancer with nonspecific signs and symptoms and very limited treatment strategies. It results in death in 15 months following diagnosis [1,2]. One of the main challenges for successful treatment of gliomas is its multiform structure [3,4]. Glioma multiforme is highly heterogeneous with different cell types and their complex interactions. Hence, it is an extremely dynamic, hierarchical microenvironment [5]. Moreover, highly activated epithelial-mesenchymal transition (EMT) and existence of glioma stem-like cells (GSCs) drive resistance and relapse in therapy [6,7,8]. Tremendously increased aggressiveness and invasiveness of GBM requires adequate single-cell tools to avoid masking rare cells and to allow quantifying heterogeneity according to morphologic, phenotypic, and functional properties of glioma cells in a population.

Since Scherer investigated distinct morphological patterns of infiltrating glioma cells in 1940, mechanophenotyping of glioma cells has been researched to identify selective biomarkers or diagnostic indicators for the incidence and prognosis of GBM [9,10]. Mechanical properties of cells, being closely linked to homeostasis of cells and their own microenvironment, have been hallmarks for defining healthy and malignant conditions of cells particularly in metastatic cancer [11,12,13,14]. Several different technologies have been developed to measure mechanical properties of cells including flow cytometry [15,16,17], atomic force microscopy (AFM) [18,19], magnetic twisting cytometry (MTC) [20], parallel-plate rheometry [11,21], optical stretching (OS) [22], optical tweezer [23,24], microfluidic ektacytometry [25,26], and micropipette aspiration [27,28]. Utilizing these modern tools for delineating mechanophenotypic properties of cells including stiffness, adhesiveness, viscosity, deformation (ratio of the area to volume), morphology, and migration trajectories of cells have been extensively investigated in cancer biology [29,30,31].

Mechanistic studies of glioma cells, emphasis on cell morphology, migration, proliferation, and invasion have been mostly interrogated in microfabricated 3D-growth chambers confined with gels that mimic the extracellular matrix (ECM). Microfabricated cell culture platforms have provided a well-controlled microenvironment for glioma cells while allowing high-resolution time-lapse microscopy imaging [32]. In addition, cell-cell adhesion strength and cell-deformability properties of glioma cells have been investigated by AFM and single cell force spectroscopy methods [33]. A pioneering work by Kaufman’s research group revealed collagen concentration-dependent growth and motility patterns of GBM cells using bulk rheology, phase-contrast, confocal reflectance, and CARS microscopy techniques [34]. Along this study, Ulrich et al. reported that ECM rigidity increased glioma motility, proliferation, and spread using time-lapse and immunofluorescence experiments [34]. Furthermore, Memmel et al. applied scanning electron microscopy (SEM) and single cell electrorotation techniques with traditional methods to characterize cell surface morphology and membrane folding of GBM cell lines [35,36].

In this study, we have investigated biomechanical properties of U87-MG (HTB-14 ™) glioma cells creating two glioma cell culturing conditions; i-U87 category: U87 glioma cells fed by the regular growth medium, and ii-U87-C category: U87 glioma cells were cultured in 50% growth medium supplemented by the 50% macrophage-depleted medium. We evaluated the behavior of glioma cells under these two growth conditions using both population-based traditional assays and microfluidic-based single-cell analysis. To characterize biomechanical heterogeneity of glioma U87 cell line at single-cell level, we measured proliferation, migration, motility, and deformation of glioma cells. We determined the expression levels of E-cadherin and Vimentin proteins between the U87 and U87-C populations according to immunostaining assays using fluorescence microscopy images.

## 2. Materials and Methods

### 2.1. Cell Culture

The U87 MG (HTB-14™) human glioma and the U937 (CRL 1593.2™) human histiocytic lymphoma monocyte cell line was purchased from ATCC (American Type Culture Collection). Cells were grown in 75 cm^2^ cell culturing flasks and 6-well plates (TTP, Switzerland at 37 °C with 5% CO_2_ (NUVE, Turkey). The U87 cells were cultured in the DMEM medium (Dulbecco’s modified Eagle’s medium, Roswell Park Memorial Institute), 10% fetal bovine serum (FBS/Sigma-Aldrich, St. Louis, MO), 1% Penicillin/Streptomycin (Pen/Strep, Pan Biotech, Germany). The U937 monocytes were maintained in RPMI 1640 medium (Roswell Park Memorial Institute, Pan Biotech, Germany), 10% FBS (Sigma-Aldrich, St. Louis, MO, USA). The human histiocytic lymphoma macrophages were differentiated from the U937 monocytes through stimulation of 3 × 10^5^ cells/mL in 5 mL RPMI 1640, 10% FBS with 5 μL working solution of 10% Phorbol 12-myristate 13-acetate (PMA, Pan Biotech, Germany) obtained from 10 ng/mL PMA/dimethyl sulfoxide (DMSO, Pan Biotech, Germany) stock solution according to standard protocols for macrophage differentiation.

The conditioned medium was harvested from U937-differentiated macrophages grown in RPMI 1640, 10% FBS for 72 h at 37 °C with 5% CO_2_. The collected media was centrifuged at 3000 rpm for 5 min on a centrifuge (Hettich-EBA-20, Germany) and filtered through a 0.2-μm filter (GVS Filter Technology, UK) and then the harvested supernatant is freshly used in the experiments.

### 2.2. Microfluidic Chip Fabrication

Microfluidic chips were designed using the CleWin 4.0 layout editor. The microfluidic cell culture platform has one inlet and outlet for cell loading and medium feeding, Figure 1. The microchamber allows cells to be cultured and visualized (1280 × 500 µm, h = 50 µm) with two types of pillars. We designed the circular (r = 90 µm, h = 50 µm) and trapezoid pillars (a = 80 µm, b = 215, h = 50 µm) to prevent polydimethylsiloxane (PDMS) from collapsing inside the cell culture microchamber and observe whether different pillar geometries effect migration of the cells. Their dimensions and pitches were determined to deform flow, slow down cells, and distribute them randomly in the cell culture microchamber [37]. To observe the imaging area by 10× objective, the distance between the pillars were determined to be 390 and 190 µm for the circular and trapezoidal ones, respectively.

The designs were patterned on a thin film chromium deposited photomask (Cr-blank) using a Vistec/EBPG5000plusES Electron Beam Lithography System. SU-8 2025 (SU-8^®^ 2025, MicroChem) was spin-coated on a four inches silicon wafer to obtain 50-μm tall structures. Next, the photoresist-coated wafers were soft baked (65 °C, 3 min and 95 °C, 5 min) and exposed to UV light (160 mJ cm^−2^, Midas/MDA-60MS mask aligner). Upon two consecutive post-baking processes (65 °C, 1 min and 95 °C, 5 min), SU-8 was developed (MicroChem’s SU-8 developer). The microfluidic chips were obtained using elastomeric polymer PDMS (Sylgard^®^ 184, Dow Corning, Midland, MI, USA) [38]. Five-mm biopsy punchers (Robbins Instruments, Chatham, MA, USA) were used to create inlet and outlet ports. The PDMS chips were irreversibly bonded both inside a 6-well plate and on a glass slide using a Corona system (BD20-AC, Electro-Technic Products Inc., Chicago, IL, USA), Figure 1.

### 2.3. Microfluidic Chip Preparation and Culturing Cells in the Microfluidic Device

To prepare the microfluidic chip, all reagents and microchips were placed into the incubator (37 °C and 5% CO_2_, NUVE, Turkey) for 30 min. Prior to cell loading, to eliminate air bubbles inside the microfluidic culture chamber, the warm medium was added using a 200-µL micropipette (Corning, New York, NY, USA). U87 glioma cells were grown as explained above (2.1 Cell culture), trypsinized (Pan Biotech, Germany), and resuspended in the DMEM medium to obtain 1.6 × 10^5^ cells/mL. Next, cell suspension with 8 × 10^4^ cells/mL was injected into the microfluidic device. Afterwards, the U87 glioma chips were mounted in the incubator (37 °C and 5% CO_2_, NUVE, Turkey) and their medium replaced with the fresh 40-µL DMEM medium every 24 h. The U87 glioma conditional (U87-C) chips were generated by replacing the regular DMEM medium with the conditioned medium when the cells were grown in the DMEM medium (37 °C and 5% CO_2_, NUVE, Turkey) overnight. The conditioned medium (explained in 2.1 Cell Culture) within microfluidic devices was also regularly refreshed once a day. Each experiment was independently performed in triplicate.

### 2.4. Cell Growth in a 12-Well Cell Culture Plate

Once the U87 cells reached 75 to 85% of confluency, cells were trypsinized (Pan Biotech, Germany) and resuspended in the fresh DMEM medium with trypan blue dye (Sigma-Aldrich, Darmstadt, Germany). Next, the number of viable cells was counted using a hemocytometer (Marienfeld, Germany). U87 cells were seeded at a density of 1 × 10^5^ cells/well in a 12-well cell culture plate (TPP, Switzerland) and allowed to adhere overnight in the incubator (37 °C and 5%, NUVE, Turkey). After overnight incubation, three wells were assigned as U87 and fed by the regular DMEM medium, while the others were assigned as U87-C and fed by the conditioned medium for five days in the incubator (37 °C and 5%, NUVE, Turkey). Both medium replacements and cell count determinations were performed for 24 h for five days. To determine the cell numbers, the cells were trypsinized (Pan Biotech, Germany), centrifuged (Eba 20, Hettich, Germany) at 1800 rpm for 10 min. The cell pellets were collected and suspended in the fresh medium with trypan blue dye. Total viable cells were counted using a hemocytometer (Marienfeld, Germany) for both U87 and U87-C growth conditions, Figure 2. Each experiment was independently performed in duplicate. Results were represented by means ± standard errors.

### 2.5. Cell Migration by Wound Healing in a 12-Well Cell Culture Plate

The wound healing assay was performed in a 6-well cell culture plate (TPP, Switzerland) where U87 cells were seeded at a density of 1 × 10^6^ cells/mL using a 2 mL DMEM medium. The prepared culture plate was kept in the incubator at 37 °C, 5% CO_2_, and allowed cells to adapt their microenvironment and adhere to the surface of the 6-well plate. Next, six wells of U87 culture were grown in the regular medium while the other six wells were maintained in the conditioned medium (Section 2.1 Cell Culture chapter) until the cells became confluent (two days). Before the scratch wound was created in the cell monolayer using a 200-µL-pipette tip (Eppendorf, Germany), phase-contrast images of the cells were observed using an inverted fluorescent microscope, the Zeiss Axio Observer (Carl Zeiss Axio Observer Z1, Germany) equipped with a 10× objective and the AxioCam Mrc5 camera. The wells were washed with the medium to remove the floating cells, 3 mL of either regular or conditioned medium per well were added into the wells, the images of the wells were acquired.

Upon 24 h of incubation, 10 μM DAPI (Life Sciences 33342) and 10 μM Propidium iodide solution (PI, Sigma-Aldrich P4864) were added into the wells containing 3 mL of medium. Next, the same microscope setting was used to acquire 24-h images of the cells to quantify the number of cells that migrated towards the wound area. The exposure rates of DAPI, FITC, and PI channels were 100, 600, and 400 mS, respectively. The excitation and emission values were 495/519 nm, long pass (LP) 515 nm for FITC, 535/617 nm LP 590 nm for PI, and 358–410 nm, LP 420 nm for DAPI, respectively. This experiment was independently performed in duplicate. The wound closure analysis was performed using ImageJ (Version 2.0 National Institutes of Health, Rockville, MD, USA). The number of migrated cells into the scratched region was manually counted using the ImageJ software. The unpaired, two-tailed Student’s *t*-test was performed to determine whether the migrated-cell number difference between the U87 and U87-C populations was significant using the GraphPad Prism 5 software.

### 2.6. Measurement of Single-Cell Migration in the Microfluidic Device

U87 cells were harvested as explained in 2.1 Cell Culture. Two separate microfluidic devices were prepared either with the regular medium or conditioned medium (as explained in 2.3 Microfluidic chip preparation and culturing cells in the microfluidic device). Cells were not stained by fluorescent dyes to eliminate the effects of labeling on biomechanical properties of cells. Next, the microfluidic chips were maintained at 37 °C, 5% CO_2_ (NUVE, Turkey). The microfluidic chips that were prepared with the regular medium were replenished with the regular medium, while the conditional medium was used to refill the conditional microfluidic devices. Microfluidic platforms were daily monitored on the microscope and images of the microchamber area were acquired using the Zeiss Axio Observer Z1 inverted microscope equipped with a 10× lens and an AxioCam Mrc5 camera. Upon imaging, the medium was replenished inside the microchambers, and microfluidic platforms were placed into the cell culture incubator. Using the obtained images, the migration distances of the cells were determined, Figure 2. The single cell migration assays in the microfluidic devices were performed for five days.

To consistently measure the migration distances of the cells, the coordinate system was defined as shown in Figure 1. The positions of the cells were defined with respect to the origin of the coordinate system. Therefore, the movement of the cells can be quantified between the inlet and outlet ports inside the microfluidic cell culture chamber for five days. Positions of single cells were manually measured using the ImageJ software (version 2.0). The velocities of cells were also calculated.

### 2.7. Immunohistochemistry

U87 cells were maintained in DMEM and U87-C glioma cells were grown in a 50% growth and 50% macrophage-depleted medium with the density of 50,000 cells/well on a round coverslip inside the 12-well plates. The cells were fixed using 4% Paraformaldehyde (Boster BioSciences, Cat No: AR1068) at room temperature for 30 min. Next, cells were permeabilized using 0.1% Triton-X100 (Sigma, T8787) and 0.1% Bovine Serum Albumin (BSA, Sigma A2058) in PBS. Upon PBS washes, coverslips were incubated overnight with primary antibody against E-cadherin (Abcam: ab1416) and Vimentin (Abcam: ab8978). Both primary antibody concentrations were adjusted to 1:100 in 2.5% BSA and 0.05% Triton X-100. The secondary antibody Alexa Fluor 488 (Thermo Fisher #A10680) was used at 1:200. DAPI (Life Sciences 33342) staining was performed after incubation. Coverslips were mounted using 50% glycerol in 1X PBS at room temperature. The coverslips were observed by a plan-apochromat 63×/1.42 Oil objective lens of a fluorescent microscope (Olympus BX60, Japan) using U-MNU2 filter with 365/10 nm excitation and 420 LP nm emission values. The exposure rates of DAPI and FITC channels were 133 µs and 175.3 µs, respectively.

The expression levels of the E-cadherin and Vimentin proteins were quantified using ImageJ (version 2.0 National Institutes of Health, Rockville, MD, USA) and GraphPad Prism 5. The number of the cells was counted using the DAPI-stained nuclei of the cells using ImageJ. Next, the total region of the acquired image area was defined to measure total fluorescence intensity from Alexa Fluor 488 dye using the ImageJ software. Afterwards, obtained row intensity density values were divided by the number of the cells. We used one-way ANOVA Tukey’s comparison test to determine whether the expression levels of E-cadherin and Vimentin proteins were significant between the U87 and U87-C glioma groups (GraphPad Prism 5).

## 3. Results

### 3.1. Influence of Conditioned Medium on U87 Proliferation in 12-Well Plate and Microfluidic Device

Two sets of glioma cells were prepared to be cultured. One of them was labeled as U87, it was grown in the regular growth medium. The other one was marked as U87-C that was cultured in the 50% growth and 50% macrophage-depleted medium (conditioned medium). Both U87 and U87-C cultures were maintained in 12-well plates for five days (see Material and Methods, Section 2.3 and Section 2.4).

We first examined the growth differences between U87 and U87-C glioma cells in culture dishes. We counted viable cells using a hemocytometer according to trypan blue staining. Figure 3a,b shows that the cell viability within the conditioned medium was similar to the growth medium (Figure 3c,d), there was no significant growth difference for 120 h according to the unpaired *t*-test (*p* = 0.407). Hence, this result validated that the phenotypic differences between U87 and U87-C were able to be further studied.

The microfabricated cell culture chamber allows proliferation and migration of the cells from the inlet port to the outlet port while enabling the monitoring behavior of cells at single-cell resolution, Figure 1. Initially, the culture medium was injected from the inlet port through the microchamber to the outlet port. The culture medium was a growth medium and conditioned medium for U87 and U87-C, respectively. Next, bubbles were removed from the microchamber, and then cell suspension was added into the inlet port. The gravity-driven laminar flow owing to 400-µm height difference between the inlet port and cell culture microchannel was obtained for gentle nutrient delivery and waste removal through the microchamber. Hence, the fluid flow did not mechanically interrupt the behavior of the cells. Figure 2c displays the number of viable cells for normal and conditioned growth environments in the microfluidic chips. As shown in Figure 2d, images of the microchannel were acquired every 24 h and the number of viable cells were manually counted. The viability was defined according to cell division via following single cells. A label-free analysis was performed to eliminate staining-induced phenotype variations.

### 3.2. Influence of Conditional Medium on U87 Cell Migration by Wound Healing Assay

The wound healing assay was performed in a 12-well cell culture plate (see Material and Methods, Section 2.5). The six wells of U87 culture were grown in the regular medium while the other six wells were maintained in the conditioned medium. The scratch wound was created in the cell monolayer using a 200-µL-pipette tip. The phase-contrast images of the wells were acquired immediately after scratching the cell monolayer and 24 h later. Next, the number of migrated cells into the scratch area was manually counted using the ImageJ software, Figure 3a–f. Each experiment was independently performed in duplicate. The U87-C group showed increased migration compared with the U87 category, however, the difference was not significant according to the Student’s *t*-test (*p* = 0.9051), Figure 3g.

Moreover, when U87 cells were cultured in the regular medium, the cells proliferated and remained on the borders of the scratched region instead of migrating to the cell-free regions. However, it was not observed for the U87-C group where U87 cells were grown in 50% DMEM and 50% macrophage depleted RPMI. Therefore, the crowdedness of the glioma cells in the central wound area was higher for the U87-C category, Figure 3.

### 3.3. Influence of Conditional Medium on U87 Cell Migration Using a Microfluidic Device

We examined the movement of the cells both in the regular medium (U87) and conditional medium (U87-C) between 48 to 120 h in a microfluidic cell culture chamber. The first 48 h of cell culture in a microfluidic device were used to allow cells to adhere to a glass surface and proliferate. Next, the images of the cells were acquired for every 12 h and analyzed as illustrated in Figure 1 (see Material and Methods, Section 2.6). Twenty cells from DMEM cultured and 20 cells from conditional- medium cultured glioma cells were selected, and their positions were recorded at 12-h intervals.

Figure 4 presents the changes of positions on the *x*-*y* axis for U87 (Figure 4a) and U87-C (Figure 4d) groups. Migration of the cells was demonstrated on the x-axes (Figure 4b,e) and on the y-axes (Figure 4c and Figure 5f). The migration distances of the U87 cells were shorter in comparison to U87-C. Movement of the U87 cell population was more uniform than the U87-C cell population on *y*-axis.

### 3.4. Influence of Conditional Medium on U87 Cell Migration Using a Microfluidic Device

Upon assessing the positions of the single cells in the microfluidic device, the area and perimeter measurements of 20 cells from U87 and U87-C populations were performed. The images of the cells were obtained every 12 h between 48 to 120 h. ImageJ was used to manually measure the diameter and perimeter of the cells. The single-cell deformation indexes (D) of each cell were calculated using Equation (1) [39,40], where π is 3.14. Figure 5 illustrates the deformation index of single cells in the microfluidic cell culture platform.
(1)$$D=1-\frac{2\sqrt{\pi \;\mathrm{Area}}}{\mathrm{Perimeter}}$$

Figure 5 elucidates that deformation indexes of glioma cells in regular medium were more heterogeneous (Figure 6a,c) in comparison to glioma cells cultured in conditioned medium (Figure 5b,d). The deformability difference between these two populations was significant according to the Student’s two-tailed t-test, *p* < 0.0001, Figure 6a. Figure 6b shows that the area to perimeter ratio of U87 population is greater than U87-C populations according to the Student’s two-tailed t-test, *p* < 0.0258.

### 3.5. Influence of Conditional Medium on the Expression of E-cadherin and Vimentin

E-cadherin and Vimentin proteins are among the molecular markers of epithelium-to-mesenchymal transition (EMT) [41]. Since, the U87-C glioma population gained more deformation (Figure 5) and migratory (Figure 6) properties in comparison with U87 glioma cells according to single-cell analysis, we evaluated the expression levels of E-cadherin and Vimentin proteins in a 12-well plate using immunostaining, Figure 7a,b. The expression levels of E-cadherin and Vimentin proteins were not significantly different between U87 and U87-C glioma cells on day 3. The weak expression level of E-cadherin significantly decreased both for U87 (*p* < 0.05) and U87-C (*p* < 0.001) populations from 3 to 5 days. Figure 7c,d displays the expression level of Vimentin protein, which was moderately weak for U87-C glioma cells and weak for U87 cells. When the expression level of Vimentin was compared between day 3 and 5, the decrease in the expression level of Vimentin was not significant for U87 glioma cells. In contrast, the decrease was significant for the U87-C glioma population (*p* < 0.001).

## 4. Discussion

The complex, dynamic, and highly heterogeneous microenvironment of glioblastoma tumors present a chicken and egg problem when we focus on understanding the interactions of glioma and immune cells. Therefore, we need to develop new methods and tools to discover important, measurable properties of cells that are not adequately measurable using traditional macroscale techniques. Elucidation of the mechanisms underlying the heterogeneity of tumor microenvironment requires quantification of cellular properties at single-cell level for a large number of cell populations and compiling the obtained results with the existing data in the literature. However, still there is a gap to be bridged between well-established, macroscale types of assay and microfluidic-based, microscale methods that match to the length and time scales of cells [42,43,44]. In this study, our aim was to investigate the influence of macrophage-secreted proteins on the behavior of glioma cells when glioma and macrophage cells were not directly in contact with each other, while integrating traditional bulk assays and microfluidic single-cell platforms.

First, we examined whether there was a significant growth difference between the U87 and U87-C glioma populations using both traditional cell culture dishes and microfabricated cell culture platforms. Figure 2 showed that there were not significant growth differences between U87 and U87-C, which allowed us to further investigate the influence of conditioned medium on biomechanical properties of U87 glioma cells. Since, the observed mechanophenotyipc differences of U87 cells might be due to macrophage-secreted proteins in the conditioned medium, not owing to growth deficiency. Although the growth of U87 cells were similar in both U87 and U87-C conditions, the micrographs of cells in Figure 2b,d showed that both in 12-well plate and microfabricated cell culture platform glioma cells were more elongated in the U87-C population. We observed that the number of rounded cells were higher for the U87 glioma population within the microfluidic chamber, where the cells adhered on the glass surface. Hence, the U87 glioma cell morphology is also dependent on the substrate stiffness, as previously reported for the LN229, LN18, and LBC3 glioma cell lines and glioma primary cells [45,46,47,48]. Moreover, our results agreed with the previous research that revealed morphological heterogeneity of U87 glioma cell line [49].

We next performed the wound healing assay using a 6-well plate. Our results presented that there was no significant difference between the number of migrated glioma cells for U87 and U87-C culture conditions (*p* = 0.9051), Figure 3g. As an important difference between U87 and U87-C glioma populations, glioma cells were distributed on the scratched region of the wound in the U87-C population compared to the U87 group where cells were more adhered to the leading edges of the wound.

To further investigate the influence of conditioned medium on the U87 cell migration, we quantified the behavior of glioma cells by cell tracking analysis. We measured the displacement of glioma cells in the microfluidic platform both for U87 and U87-C populations, Figure 4. Our results verified that glioma cells moved longer distances with a relatively high migration speed in the U87-C population, when cells were fed by the conditioned medium, Figure 4a–d. Both U87 and U87-C cell populations disseminated more on the *y*-axis. Movement of the cells along the *x*-axis were more uniformly distributed. Herein, neither U87 nor U87-C populations exhibited directionality in their movement. Still, these results show that GBM cells exhibited high heterogeneity in migration displacement, orientation, and velocity [49,50]. To the best of our knowledge, our study presents for the first-time single cell tracking analysis of glioma cells for 120 h in the microfluidic platform. Mostly, migration and morphology assays have been performed for shorter time frames (0–10 h) in vitro assays [35,47].

Afterwards, we assessed whether indirect contact between glioma and macrophage cells influences deformation capability of glioma cells. We determined deformation indexes (DI) of glioma cells according to area and perimeter measurements of glioma cells. Figure 5 shows that glioma cells in the U87-C population have significantly higher deformation indexes (DI > 0.8) in comparison to the U87 glioma population (DI < 0.4), *p* < 0.0001. Additionally, glioma cells displayed higher deformation heterogeneity in the regular growth medium in comparison to the conditioned medium.

Taken together, culturing glioma cells in 50% DMEM and 50% macrophage-depleted medium influenced morphology, motility, and deformation of glioma cells. To evaluate whether these altered mechanical phenotypes were linked to epithelium-to-mesenchymal transition, we assessed the expression levels of E-cadherin and Vimentin proteins both in U87 and U87-C populations. E-cadherin and Vimentin proteins are among the molecular markers of epithelium-to-mesenchymal transition (EMT) [41]. The EMT process introduces a malignant phenotype, spindle-shaped morphology, and metastatic functions for cancer cells by altering the activation of transcription factors, expression levels of specific microRNAs and cell-surface proteins, as well as organization of cytoskeletal proteins. Epithelial cells become mesenchymal with low levels of E-cadherin and high levels of vimentin expressions. Our results indicated that the indirect effect of macrophages under our experimental conditions did not provide mesenchymal phenotype to U87 glioma cells.

## 5. Conclusions

This study demonstrated a single-cell mechanophenotyping approach for U87 glioma cells while integrating microfluidic cell culture platforms with macroscale traditional assays. We cultured glioma cells either in the regular growth medium, denoted as U87 cell population, or in the 50% regular growth medium and 50% macrophage-depleted medium (conditioned medium), referred to as U87-C. We cultured both U87 and U87-C glioma populations in traditional cell culture dishes and microfluidic platforms for five days. We quantified proliferation, morphology, motility, migration, and deformation properties of glioma cells using single-cell analysis, which are directly linked to biomechanical features of cells. Our results presented that there was no significant growth difference between U87 and U87-C glioma populations, however U87-C glioma cells exhibited a slightly weaker proliferation in the microfluidic device. U87 and U87-C glioma populations were morphologically heterogeneous both in the bulk and microfluidic assays. We observed that the macrophage-conditioned stimulation provided glioma cells with slightly increased motility and extremely significant deformation capabilities (*p* < 0.0001). We measured the expression levels of E-cadherin and Vimentin proteins to assess whether the phenotype of glioma cells in the U87-C population were transformed into the mesenchymal phenotype. However, immunostaining experiments verified that observed phenotypic changes were statistically E-Cadherin and Vimentin independent. Our results confirmed enormous heterogeneity of U87 glioma cell line in terms of mechanophenotypic properties. Herein, we integrated microscale and macroscale growth conditions and quantified mechanophenotypic properties of glioma cells thanks to the microfluidic cell culture platform. Considering our results, integration of microfabricated, single-cell level and traditional, population-level assays will be the future of modern techniques in cell biology.

## Figures and Tables

**Figure 1 micromachines-11-00845-f001:**
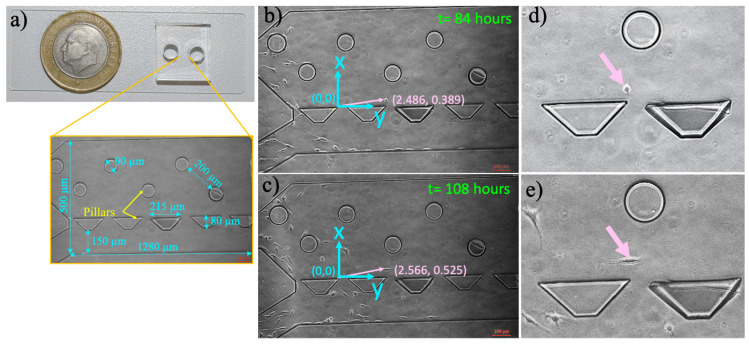
Microfluidic cell culture platform and measurement of single-cell migration in the microfluidic device. (**a**) The polydimethylsiloxane (PDMS) microfluidic chamber on a glass slide, (**b**) the micrographs of microfluidic chamber with pointed pillars and dimensions. (**c**) The blue line indicates the position of the coordinate system (0, 0) with *x*- and *y*-axis. The pink arrow points to the position of a cell according to origin, (**b**) x: 2.486, y: 0389 at 84 h, (**c**) x: 2.566, y: 0.525 at 108 h, (**d**) and (**e**) demonstrate the zoomed images of this cells in (**b**) and (**c**), respectively. The scale bar shows 100 µm.

**Figure 2 micromachines-11-00845-f002:**
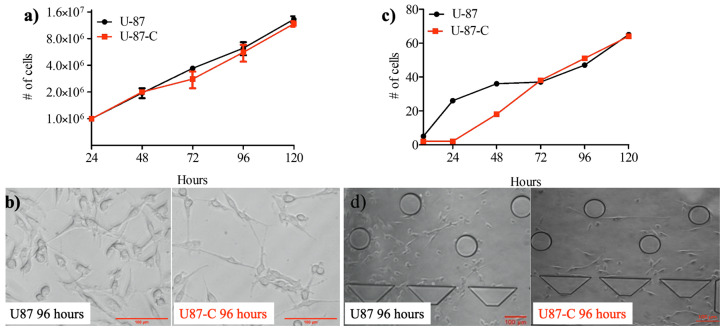
Growth comparison. Glioma cells were grown in DMEM medium (U87) and in 50% DMEM and 50% macrophage-depleted medium (U87-C) in the 6-well culture dish and microfluidic platform. (**a**) The number of viable cells for five days in a 6-well plate, (**b**) the micrographs of U87 and U87-C cells for a 96-h growth. The scale bar shows 20 µm. (**c**) The number of viable cells for five days in the microfluidic device, (**d**) micrographs of glioma cells for the 96-h growth. The scale bar shows 100 µm. The number of cells present the mean ± standard error for two independent experiments.

**Figure 3 micromachines-11-00845-f003:**
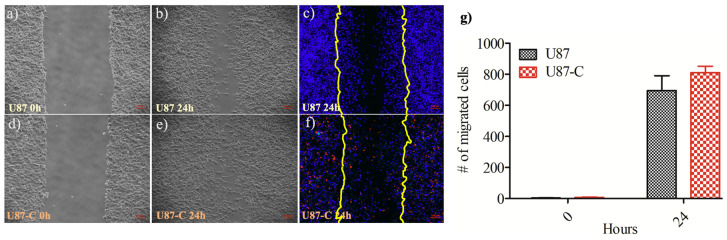
Analysis of U87 and U-7-C cells migration by the in vitro wound-healing assay. The phase images of U87 cells when the (**a**) wound created at 0 h, (**b**) phase images of wound closure at 24 h, (**c**) fluorescence images of wound closure at 24 h, the nucleus of the cells are labeled with DAPI and displayed in blue, dead cells are PI-stained and shown in red, yellow lines present the wound area created at 0 h. The same settings were applied for U87-C (**d**–**f**). The images acquired with 10× magnification; the scale bar shows 100 µm. (**g**) The number of migrated cells at 0 and 24 h. The results represent the mean ± standard deviation of two independent experiments. There was no significant difference according to the Student’s unpaired, two-tailed t-test, *p* = 0.9051.

**Figure 4 micromachines-11-00845-f004:**
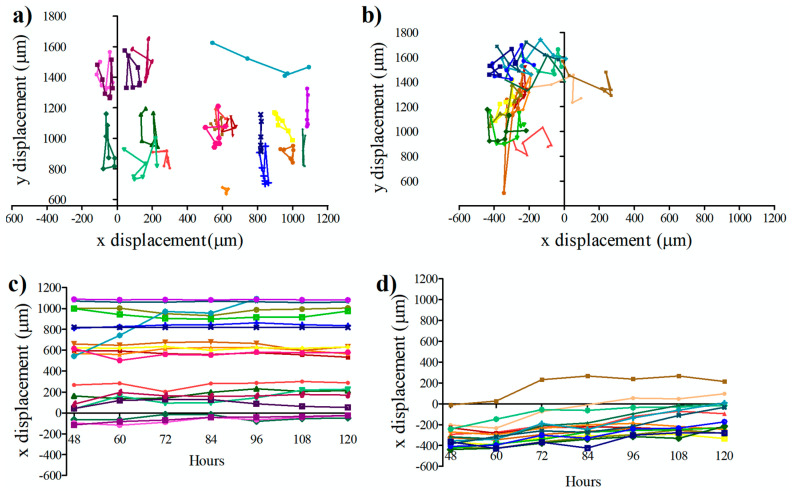
Migration of single cells in the microfluidic cell culture chamber. Coordinates of the cells in the microfluidic cell culture device were measured every 12 h between 48 to 120 h. Movement of the U87 cells (**a**) at x-y axes (**b**) on the *x*-axis, (**c**) on the *y*-axis. Movement of the U87-C cells (**d**) at *x*-*y* axes, (**e**) on the *x*-axis, (**f**) on the *y*-axis. The number of analyzed cells for each group is 20. U87 indicates that cells cultured in DMEM medium, U87-C defines that cells were grown in 50% DMEM and 50% macrophage-used RPMI medium. Each color represents the single cells and color coding was consistent in each group.

**Figure 5 micromachines-11-00845-f005:**
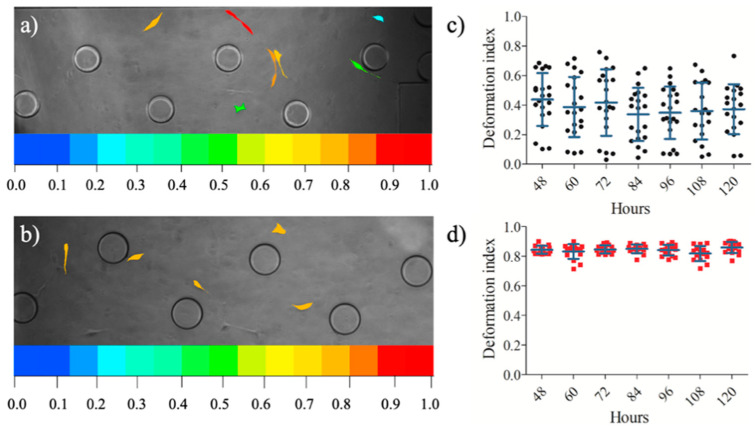
Deformation indexes of the cells in the microfluidic device. The phase images of the cells with a colorimetric deformation scale, the range of deformability from coolest colors (blue: 0) to warm colors (red:1) represent enhanced deformability indexed (**a**) U87 population, (**b**) U87-C population. The deformation indexes of 20 glioma cells between 48- and 120-h (**c**) for U87 population, (**d**) for U87-C population. The data presents the mean ± standard error.

**Figure 6 micromachines-11-00845-f006:**
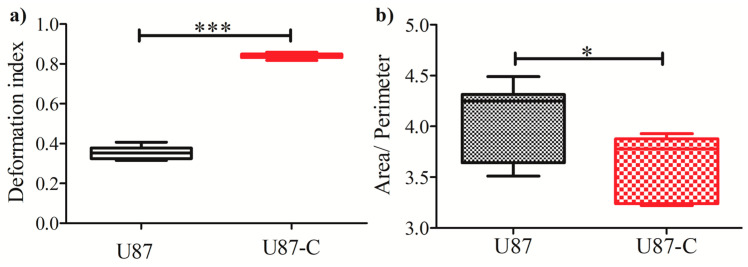
Comparisons of deformation indexes and area/perimeter. (**a**) Deformation indexes differences between U87 and U87-C. Student’s two-tailed *t*-test was applied, *p* < 0.0001. (**b**) Differences of the area to perimeter ratio according to the Student’s two-tailed t-test, *p*-value is 0.0258. * and *** implies for *p* < 0.05 and *p* < 0.0001, respectively.

**Figure 7 micromachines-11-00845-f007:**
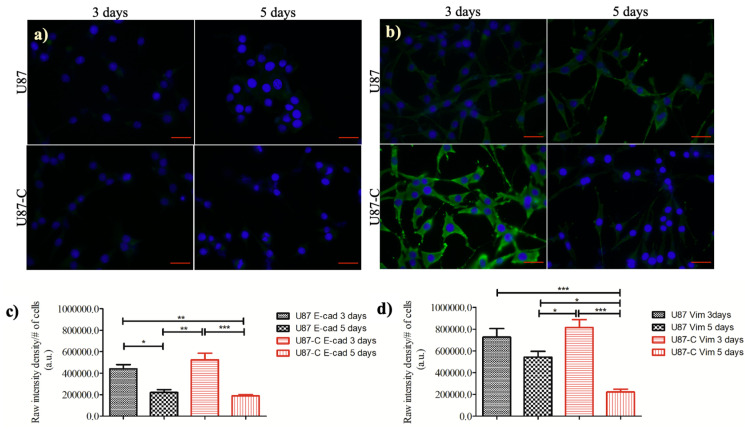
Comparisons of E-cadherin and Vimentin expressions. (**a**) Immunofluorescence staining of E-cadherin (E-cad), (**b**) Vimentin (Vim) proteins (green) with nuclei counterstained (blue) by DAPI. The scale bar is 25 µm, the magnification is x63. Quantification of (**c**) E-cadherin and (**d**) Vimentin expressions of U87 and U87-C glioma cells for 3 and 5 days. The one-way analysis of variance Tukey’s multiple comparison test was applied. *, **, and *** denotes for *p* < 0.05, 0.01, and *p* < 0.0001, respectively.

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
