# Peer review of "Single-Cell Mechanophenotyping in Microfluidics to Evaluate Behavior of U87 Glioma Cells"

_micromachines, 2020, doi:10.3390/mi11090845_

Round 1
Reviewer 1 Report
Authors have used traditional and microfluidic-based methods to study the U87 glioma cell behavoir in two different media quantifying the deformation of the cell for the first time. They showed reasonable agreement between the two methods. The work is systematically presented and would be potentially interesting for the readers of the journal.
Below are some minor comments a major comment about the microfluidic design.
Although the message is clear. The language needs some improvement in English
Be consistent in using single-cell (single cell). Replace all “single cell”s with “single-cell” when necessary.
title of cited authors should not be used in a research article (e.g. line 37 use Scherer instead of Dr. Scherer)
include other microfluidic mechano phenotyping microfluidic devices/methods that were and can be used single-cell/ cancer cells analysis in the paragraph starting at line 37 e.g. in the interaction sections. E.g. Akbaridoust et al. Meas. Sci. Technol. 2018, Henry et al. Sci Transl. Med. 2013, Akbaridoust et al. Exp. Fluids. 2018
Lin47 Either use area-to-volume ratio or ratio of the area to volume
Line 94 The circular and trapezoid pillars (r = 90 um, h = 50 um) “prevent” … (not avoid)
Explain the reason for using pillars in the microfluidic chip fabrication section. Are the pillars are used only to avoid collapsing PDMS? A channel with a width of 500 microns and a height of 50 microns would not collapse without the pillars There are many articles with wider and higher channel depth (e.g. Szydzik et al. Anal.Chem 2019, Akbaridoust et al. Biomicrofluidics 2020). What is the reason for using both trapezoid and circular pillars?
The arrows and number in figure 1c are misaligned. The aspect ratio of figure 1c is not correct (circles look like ellipses). Figures 1c, 1b, and ac are all the same thing and repetitive. Everything can be shown in figure 2b with a larger field of view.
The gravity-driven laminar flow was obtained for gentle nutrient delivery and waste removal through the microchamber. Hence, the fluid flow did not mechanically interrupt behaviour of the cells. gravity-driven laminar in microfluidic devices can be very fast and create high strain rates. Have you done any calculation/experiment to calculate the flow strain/shear rate and compare it with the cell stiffness (from the literature)? What about the impact of the pillars?
Author Response
Answers to Reviewers
We would like to inform that we greatly appreciated our reviewer’s valuable effort and time to make our work published in prestigious Micromachines journal.
Reviewer_1: Authors have used traditional and microfluidic-based methods to study the U87 glioma cell behavoir in two different media quantifying the deformation of the cell for the first time. They showed reasonable agreement between the two methods. The work is systematically presented and would be potentially interesting for the readers of the journal.
Below are some minor comments a major comment about the microfluidic design.
Reviewer_1 #1: Although the message is clear. The language needs some improvement in English
Answer to Reviewer 1: We agree with our reviewer and we improved the scientific language.
Reviewer_1 #2: Be consistent in using single-cell (single cell). Replace all “single cell”s with “single-cell” when necessary.
Answer to Reviewer 1: Thanks for this notice, we corrected.
Title: Single-cell mechanotyping
Line 164: single-cell migration
Line 293: single-cell deformation
Reviewer_1 #3: Title of cited authors should not be used in a research article (e.g. line 37 use Scherer instead of Dr. Scherer)
Answer to Reviewer 1: Thank you very much, we corrected it as follows.
Since Scherer investigated distinct morphological patterns of infiltrating glioma cells in 1 940,
Reviewer_1 #4: include other microfluidic mechano phenotyping microfluidic devices/methods that were and can be used single-cell/ cancer cells analysis in the paragraph starting at line 37 e.g. in the interaction sections. E.g. Akbaridoust et al. Meas. Sci. Technol. 2018, Henry et al. Sci Transl. Med. 2013, Akbaridoust et al. Exp. Fluids. 2018
Answer to Reviewer 1: We emphasized the mechanophenotyping microfluidic devices/methods those have been implemented for the single-cell/ cancer cells analysis in the Introduction Section (Line 37), thank you very much for sharing these studies with us. We included “Henry et al. Sci Transl. Med. 2013”. However, the other references were not directly related to this work.
Line 39-41: Mechanical properties of cells, being closely linked to homeostasis of cells and their own microenvironment, have been hallmarks for defining healthy and malignant conditions of cells particularly in metastatic cancer [11-14].
- Tse, H.T.; Gosset, R. D.; Moon, Y. S.; Masaeli, M.; Sohsman, M.; Ying, Y.; Mislick, K.; Adams, P.; Rao, J.; Di Carlo, D. Quantitative Diagnosis of Malignant Pleural Effusions by Single-Cell Mechanophenotyping. Science Transitional Medicine 2013, 5(212), 212ra163. doi: 10.1126/scitranslmed.3006559
Reviewer_1 #5: Lin47 Either use area-to-volume ratio or ratio of the area to volume
Answer to Reviewer 1: Thanks, corrected as: viscosity, deformation (ratio of the area to volume),
Reviewer_1 #6: Line 94 The circular and trapezoid pillars (r = 90 um, h = 50 um) “prevent” … (not avoid)
Answer to Reviewer 1: Thank you very much and please accept our apologies for these language-based corrections.
The circular and trapezoid pillars (r = 90 µm, h = 50 µm) prevent polydimethylsiloxane (PDMS) to
Reviewer_1 #7: Explain the reason for using pillars in the microfluidic chip fabrication section. Are the pillars are used only to avoid collapsing PDMS? A channel with a width of 500 microns and a height of 50 microns would not collapse without the pillars There are many articles with wider and higher channel depth (e.g. Szydzik et al. Anal.Chem 2019, Akbaridoust et al. Biomicrofluidics 2020). What is the reason for using both trapezoid and circular pillars?
Answer to Reviewer 1: Yes, we used pillars both to prevent collapse and to observe whether their geometry affects migration of the cells (including circular and trapezoid ones), however, our observations showed that the shape of the pillars did not affect on migration of the cells. Thanks.
Reviewer_1 #8: The arrows and number in figure 1c are misaligned. The aspect ratio of figure 1c is not correct (circles look like ellipses). Figures 1c, 1b, and ac are all the same thing and repetitive. Everything can be shown in figure 2b with a larger field of view.
Answer to Reviewer 1: Thank you very much for your advice. We corrected it as you recommended. Page: 4, Lines: 175-182.
Figure 1. Microfluidic cell culture platform and measurement of single-cell migration. a) PDMS microfluidic chamber with pointed pillars and dimensions. The blue line indicates the position of the coordinate system (0,0) with x- and y-axis. The pink arrow points the position of a cell according to origin, b) x: 2.486, y: 0389 at 84 hours, c) x: 2.566, y: 0.525at 108 hours, d) and e) demonstrate the zoomed images of these cells in b) and c), respectively. The scale bar shows 100 µm.
Reviewer_1 #9: The gravity-driven laminar flow was obtained for gentle nutrient delivery and waste removal through the microchamber. Hence, the fluid flow did not mechanically interrupt behaviour of the cells. gravity-driven laminar in microfluidic devices can be very fast and create high strain rates. Have you done any calculation/experiment to calculate the flow strain/shear rate and compare it with the cell stiffness (from the literature)?
Answer to Reviewer 1: Thank you very much for asking this important explanation. Basically, the medium flow was generated due to the potential difference between the inlet port and the microchannel owing to 400 um height difference. Therefore, we did not encounter fast flow rate or high shear stress in the microchannel (as you stated, which can be the problem for gravity-driven flow). However, this is a very important point that we have revised upon your comment as follows: Lines 236-239:
The laminar flow owing to 400-µm height difference between the inlet port and the cell culture microchannel was obtained for gentle nutrient delivery and waste removal through the microchamber. Either the fluid flow or the pillars did not mechanically interrupt behavior of the cells.
Reviewer_1 #10: What about the impact of the pillars?
Answer to Reviewer 1: The pillars did not affect the migration or deformation of the cells. Besides, the single-cell analysis has been performed within the area where there were not pillars in the microfluidic chamber. The pillars helped us to measure positions of the cells with respect to coordinate system that we located at the right corner of the first trapezoid pillar in the microchannel.
We greatly appreciated our Reviewers very important contribution.
We thank for our Reviewer’s kind help, valuable time and careful revision.
Reviewer 2 Report
The authors fabricated a microfluidic cell culture platform to investigate the indirect influence of macrophages on glioma cell behaviour. Through a series of experiments, they quantified proliferation, morphology, motility, migration, and deformation properties of glioma cells at the single-cell level and compared the results with standard cell culture dishes. It is a very well-written manuscript, and the results have presented very clearly. However, I believe the authors could have fabricated a more novel microfluidic platform to study single-cell migration and deformation. For instance, the chip does not have proper labelling, making single-cell tracking very difficult. I wonder how they dynamically located the x-y locations of a specific cell over time (i.e., Figure 5). Also, by properly designing the microfluidic chip, they could have investigated the invasion assay in an in vivo-like microenvironment (e.g., injecting hydrogel in one side of the chip). As such, it would be better that authors more explain the advantages of using their microfluidic cell culture.
Further comments:
- Please explain more about the engineering of the pillars and the spacing. Why there are two different geometries (circular and trapezoidal ones) and how you chose the dimensions and pitch?
- Figure 3: The labelling of parts b and c are wrong on the figure. Also, in the caption, the authors talk about “6-well” culture dish instead of “12-well” one.
Author Response
Answers to Reviewers
We would like to inform that we greatly appreciated our reviewer’s valuable effort and time to make our work published in prestigious Micromachines journal.
Reviewer 2: The authors fabricated a microfluidic cell culture platform to investigate the indirect influence of macrophages on glioma cell behaviour. Through a series of experiments, they quantified proliferation, morphology, motility, migration, and deformation properties of glioma cells at the single-cell level and compared the results with standard cell culture dishes. It is a very well-written manuscript, and the results have presented very clearly.
Reviewer_2 #1: However, I believe the authors could have fabricated a more novel microfluidic platform to study single-cell migration and deformation. For instance, the chip does not have proper labelling, making single-cell tracking very difficult. I wonder how they dynamically located the x-y locations of a specific cell over time (i.e., Figure 5).
Answer to Reviewer 2: We agree with our reviewer, however, due to economic reasons (budget of the project) we cannot afford cleanroom expenses. Therefore, we used the microfluidic platform that we have already made to generate the most quantitative data that we can obtain. We are sorry for this inconvenience and sharing it with our reviewers.
Here, we always located the x-y coordinate frame (Figure 1b, 1c) at the right corner of the first trapezoid pillar in the microchannel (as shown in Figure 1) and manually measured positions of the cells according to its origin and calculated the migration distance at each time point.
Lines: 185-186: To consistently measure the migration distances of the cells, the coordinate system was defined as the right corner of the first trapezoid pillar in the microchannel (as shown in Figure 1).
Another point that bothers us is we cannot make real-time observations of the cells on the microscope since we do not have an incubator (chamber) on the microscope that we can maintain cellular growth at 37ËšC with humidity. Hence, we take the snap shots of cells in a way that we did not disturb growth of the cells at room temperature on the microscope. Please accept our apologies for mentioning the facts that limits our research. Thank you very much for asking and recommending a novel design that we can track single cells and study not only cell migration, deformation but also adhesion.
Reviewer_2 #2: Also, by properly designing the microfluidic chip, they could have investigated the invasion assay in an in vivo-like microenvironment (e.g., injecting hydrogel in one side of the chip). As such, it would be better that authors more explain the advantages of using their microfluidic cell culture.
Answer to Reviewer 2: We completely agree with our Reviewer 2. Due to the mentioned points in the first question/comment, we cannot make it for now. However, we consider these suggestions for further studies. We thank our Reviewer.
Further comments:
Reviewer_2 #3: Please explain more about the engineering of the pillars and the spacing. Why there are two different geometries (circular and trapezoidal ones) and how you chose the dimensions and pitch?
Answer to Reviewer 2: Thanks. We explain the pillars better and provided the reference [37] for better explanation. Lines: 94-100.
We designed the circular (r = 90 µm, h = 50 µm) and trapezoid pillars (a = 80 µm, b= 215, h = 50 µm) to prevent polydimethylsiloxane (PDMS) collapsing inside the cell culture microchamber and observe whether different pillar geometries effect migration of the cells. The dimensions and pitches of the pillars were determined to deform flow, slow down cells and distribute cells randomly in the cell culture microchamber [37]. To observe the imaging area by 10 x objective, the distance between the pillars were determined to be 390 µm and 190 µm for the circular and trapezoidal ones, respectively.
- Texier, B. D.; Laurent, P.; Stoukatch, S.; Dorbolo, S. Wicking through a confined micropillary array. Microfluid Nanofluid 2016, 20 (53). Doi: 10.1007/s10404-016-1724-3
Reviewer_2 #4: Figure 3: The labelling of parts b and c are wrong on the figure. Also, in the caption, the authors talk about “6-well” culture dish instead of “12-well” one.
Answer to Reviewer 2: Thank you very much for your careful revision. We both corrected the labeling in Figure 2 and the title 2.5 (Line: 141) as follow.
2.5. Cell migration by wound healing in 12-well cell culture plate
Figure 2. Growth comparison. Glioma cells were grown in DMEM medium (U87) and in 50% DMEM and 50% macrophage-depleted medium (U87-C). The number of viable glioma cells a) in the 12-well culture dish with b) the micrographs of the cells for 96-hour growth. c) The number of viable glioma cells in the microfluidic platform with d) the micrographs of the cells for 96-hour growth. The scale bar shows 100 µm. The number of cells presents mean ± standard error for two independent experiments.
We greatly appreciated our Reviewers very important contribution.
We thank for our Reviewer’s kind help, valuable time and careful revision.
Round 2
Reviewer 1 Report
The authors have carefully attempted to address the questions and revised the manuscript (please annotate your manuscript when you revise it). It can be accepted.